# Factors Impacting COVID-19 Vaccine Uptake and Confidence Among Immigrant and Refugee Populations in Canada

**DOI:** 10.3390/ijerph22040493

**Published:** 2025-03-26

**Authors:** Ilene Hyman, Ayesha Khan, Iwo Effiong

**Affiliations:** 1Dalla Lana School of Public Health, Toronto, ON M5T 3M7, Canada; 2Canadian Association of Community Health Centres, Toronto, ON M5T 3A9, Canada; akhan@cachc.ca (A.K.); ieffiong@cachc.ca (I.E.)

**Keywords:** vaccine uptake, immigrants, refugee, access to healthcare, COVID-19

## Abstract

Objective: This study examines the barriers and facilitators to COVID-19 vaccination among immigrant and refugee populations, with a focus on informing primary healthcare stakeholders on effective strategies to address the health needs of these groups. Although conducted in Canada, the findings are relevant to countries facing similar challenges in promoting vaccine uptake among migrant communities. Methods: As part of an evaluation of best practices in COVID-19 vaccination promotion and provision, data were collected using in-depth key informant interviews with a cross-section of primary care stakeholders (n = 11). Main findings: Key barriers to vaccine promotion and provision included distrust of health and government services, misinformation, lack of vaccine confidence, and access or systems-level barriers. Effective facilitators were relationship-building and equity-driven approaches, such as community engagement and development, culturally and linguistically effective communication, one-on-one supports, and collaboration with community members as valued partners and staff. These strategies were identified as best practices that enhanced vaccine confidence and uptake. Conclusion: The risk and impacts of COVID-19 are disproportionately distributed worldwide, affecting migrant populations in many countries. Primary healthcare stakeholders must understand the barriers and facilitators to vaccine promotion to effectively address health inequalities. Increasing vaccine uptake and confidence among immigrant and refugee populations requires targeted and tailored approaches that are culturally responsive and equity-informed. These findings provide valuable insights for health systems globally, supporting efforts to reduce health inequities by using inclusive vaccination strategies.

## 1. Introduction

The COVID-19 pandemic exposed and exacerbated significant social and health inequities. Globally, it is well-documented that the risk and burden of COVID-19 is disproportionately higher among equity-deserving populations [1,2,3,4,5]. In Canada, the term “equity-deserving populations” refers to groups of people who, because of systemic discrimination, face barriers that prevent them from having the same access to the resources and opportunities that are available to other members of society and that are necessary for them to attain just outcomes [6]. Nationally and internationally, much attention has focused on reducing COVID-19 inequities in hard-to-reach or marginalized population groups who face a myriad of individual, social, and economic barriers to COVID-19 vaccine uptake [7]. The World Health Organization recommends that vaccine prioritization within countries should include refugees and migrants [8].

According to Canada’s most recent population census in 2021, more than 8.3 million people, or almost one-quarter (23.0%), of the Canadian population were, or had ever been, a landed immigrant or permanent resident in Canada. This is among the highest proportion of all G7 countries [9]. In the previous century, the majority of immigrants in Canada came from Europe; however, over the past 50 years, the share of new immigrants from Europe has declined, and the share of new immigrants born in Asia (including the Middle East) has been increasing. The wide variety of source regions of immigrants contributes to the linguistic diversity in Canada. In 2021, 69.4% of recent immigrants did not report having English or French as their mother tongue. A refugee is someone living outside their home country that is unable or unwilling to return to their home country because of a justifiable fear of persecution. As of 2022, refugees accounted for approximately 17% of all permanent residents in Canada [10]. Mexico was the leading source country for refugee claimants, followed by Haiti, Turkey, Colombia, and Iran.

While this paper examines access to COVID-19 vaccination among immigrants and refugees in Canada as a whole, it is important to recognize that these groups have distinct migration trajectories and post-migration experiences that uniquely influence health access and outcomes. Furthermore, the diversity within immigrant and refugee populations significantly affect health outcomes and necessitate tailored approaches to healthcare delivery. The concept of “social determinants of health” (SDOH) acknowledges that health is largely shaped by social and economic factors, requiring health promotion and policies that go beyond biomedical and behavioral risk factor approaches [11,12]. Since SDOH may be distributed differently and have differential impacts on different population groups, the need to consider intersectionality is critical [13]. Immigrant population sub-groups at high risk of transitioning to poor health include seniors, low-income immigrants, and recent immigrants who are members of a minority racial group [14,15,16]. Refugees who are often dealing with the aftermath of trauma and adverse living conditions [17,18,19], and who are less likely than other categories of migrant women to be proficient in their host country’s language [20,21], are also at risk of poor health outcomes.

International and national health research indicate that inequities observed in COVID-19 incidence, morbidity, and mortality require *increased* COVID-19 vaccination and prioritization of immigrant and refugee populations [22,23]. Vaccination is a proven and highly effective strategy to reduce morbidity and mortality from COVID-19; however, intentions to receive a COVID-19 vaccine have been steadily declining, while rates of vaccine hesitancy are increasing [24]. The rollout of COVID-19 vaccinations in Canada, led by the federal and provincial/territorial governments in collaboration with public health agencies, began in December 2020, by prioritizing vaccination based on health and age risk factors. Vaccinations were primarily delivered through centralized public health-led mass vaccination clinics, pharmacies, and mobile units, often excluding the participation of the primary healthcare sector as a principal partner [25]. In 2022, 86% of Canadians reported receiving primary healthcare from family physicians and nurse practitioners in private practices, community-based clinics, or community health centers [26]. As of December 2021, approximately 76% of the Canadian population completed their two-dose primary series of COVID-19 vaccinations; however, as of March 2023, only 19% of the total population either completed their primary series or received a booster dose within the last 6 months [27,28]. Canada is currently rolling out a 7th COVID-19 vaccination, which is recommended for people who are at increased risk of infection or severe illness [29].

New and innovative strategies are required to increase vaccine confidence and uptake among immigrant and refugee populations, and they involve the primary healthcare sector as a partner since Canadians are more likely to trust their health professionals than public health officials [30,31].

The objective of this paper is to share findings on key barriers and facilitators to COVID-19 vaccination among immigrant and refugee populations across Canada and to make recommendations to researchers, public health professionals, and healthcare planners on ‘best practice’ strategies to improve COVID-19 vaccine uptake among immigrants, refugees, and other equity-deserving population groups.

## 2. Materials and Methods

This evaluation employed qualitative interviews to investigate the barriers and facilitators to COVID-19 vaccine uptake among equity-deserving populations. The target population included a sample of primary care stakeholders working primarily in Community Health Centers (CHCs) across Canada. CHCs in Canada are non-profit organizations that integrate primary healthcare, health promotion programming, and social services through a collaborative tea-based framework to support the health and wellbeing of the diverse communities they serve. CHCs fill the gaps of primary healthcare and health promotion to address the high priority needs of their clients and community to remove barriers to healthcare and social service access [32,33,34]. As of the latest data, there are approximately 300 registered Community Health Centers (CHCs) in Canada [35].

The sample was drawn from respondents to a national cross-sectional survey used to evaluate the vaccine promotion and provision activities in Community Health Centers (CHCs). There were 77 CHCs who responded to the survey. From the sampling frame of respondents, we selected 11 primary care stakeholders to participate in in-depth interviews in consideration of COVID-19 promotion and provision activities, equity-deserving populations served, national geography, and rural/urban setting. Among the 11 key-informants selected, all were engaged in vaccine promotion activities with equity-deserving populations. Two were located in Atlantic Canada, five in Central regions, and four in Western Canada. Four CHCs served rural populations.

Each primary care stakeholder was told the purpose of the interview, and permission was requested to participate in a recorded online interview. Ethical review and approval were not required for this study, as it was conducted as part of a program evaluation rather than research as defined by the Tri-Council Policy Statement: Ethical Conduct for Research Involving Humans—TCPS 2 (2022). The evaluation used a voluntary national survey and voluntary key informant interviews with a low-vulnerability population of primary care stakeholders (e.g., managers, decision-makers, and clinical leads).

Interviews were set at a mutually convenient time and conducted by at least of the three investigators. Participants were provided with a copy of the interview questions in advance (see Appendix A). The interviews took an average of 1 h to conduct. Each interview was recorded and a transcript was produced. Braun and Clarke’s [36] 6-step thematic analysis framework was used to guide data analysis. This structured approach allows for researchers to systematically identify patterns and meanings within qualitative data, providing a robust foundation for drawing conclusions and making interpretations. Two separate investigators reviewed and coded each interview. Findings related to the barriers and facilitators specific to immigrant and refugee populations were extracted. An intersectional analysis was applied to identify findings specific to gender, racial, and other social identities.

## 3. Results

Eleven primary care stakeholders participated in the in-depth interviews. This section presents the findings related to the barriers and facilitators of COVID-19 vaccination.

### 3.1. Barriers to COVID-19 Vaccination

Primary care stakeholders identified complex individual and systemic barriers to improving COVID-19 vaccination rates among the equity-deserving populations they serve (Table 1).

Several stakeholders reported that a history of injustice leading to a longstanding distrust of health and government services was a significant reason for low COVID-19 vaccine uptake among equity deserving populations, notably African, Black, and Caribbean populations. Healthcare providers were faced with addressing apprehension and resistance embedded within communities that contributed to low vaccine confidence and acceptance.

Another key barrier was misinformation. Primary care stakeholders reported that their clients did not always have access to appropriate and evidence-based information and, as a result, there was a lower perceived severity of the disease.


*They didn’t have the same information, they weren’t watching the news like everyone else, they weren’t understanding the gravity of the situation, the same that we had privy to, and the privilege to receive and so it was having those conversations all the time.*


Lack of vaccine confidence in the population was another frequently cited barrier. Numerous examples were provided of clients who expressed doubts about vaccine safety, fears of adverse events, and COVID-19 fatigue in the populations they served. Access barriers that prevented equitable access to vaccination services included lack of transportation, cost, and distance to vaccination site. In some rural areas, people had to travel an hour and a half to access vaccine services.

Several barriers could be described as systemic, as they were associated with insufficient resources or operational challenges. For example, a lack of access to primary care in catchment populations impacted COVID-19 vaccine confidence, uptake, and access.


*We have 10% of the population without a primary care provider. One of our challenges as a CHC, and serving a large population of newcomers, is to bridge care as arrivals come in. We’ve had people for years now because there’s nowhere else to send them. We are in a very challenging situation in terms of primary care overall for the municipality.*


Primary care stakeholders identified inequities in the distribution and impact of social determinants of health that impacted their clients. Social determinants of health (SDOH) refer to the conditions in which people live and work including, early childhood development, employment and working conditions, income and its equitable distribution, food security, education, social exclusion, and social safety nets. Clients prioritized non-medical factors such as basic housing and food security needs over COVID-19 vaccination, which presented challenges to primary care providers when engaging in COVID-19 vaccine promotion, provision, and outreach. Stakeholders also discussed how provincial and regional health authorities did not always consider the impacts of the social determinants of health when implementing their vaccine rollouts, and these factors have a direct influence on an individual’s health and wellbeing.


*I felt like I was fighting the entire (time) for authorities to understand that when you have a person who doesn’t have a place to live, they can’t go anywhere. If they’re symptomatic, doors are closed, they need to wear masks, but people that don’t have the resources can’t do any of those things.*


### 3.2. Facilitators to COVID-19 Vaccination

Primary care stakeholders identified facilitators to vaccine uptake and acceptance (Table 2). These facilitators were often not COVID-19 specific, in that they included relational models and approaches to working with their clients who were from equity-deserving communities.

Several stakeholders attributed successful vaccine promotion and provision efforts to the trusted and longstanding relationships they had with clients and their communities.


*If you build those trusting relationships in normal times, when crisis comes, people have rapport and trust with you.*


Other stakeholders described the use community engagement and strength-based approaches to effectively engage and build trust with community members, including hiring immigrant community members and people with lived experience. Viewing community as the ‘expert’ and involving community members/elders in decision-making and program development was viewed as vital to the effectiveness of vaccine promotion and provision.


*Our biggest intervention that was the most successful… was harnessing the unrecognized leadership of community residents… and community ambassadors and arming them with information and resources.*


In some jurisdictions, the reach and uptake of vaccination services was expanded by using peer ambassadors, trained and trusted community members, who engaged in culturally and linguistically accessible COVID-19 vaccine outreach and promotion to reach and support their communities. In Ontario, funding was provided to lead agencies (mostly CHCs) to engage and hire community ambassadors. The advantages of working with trained peer ambassadors, in addition to healthcare professionals, is evident in the following quote:


*Getting the right people involved is my biggest piece of advice. We have lots of navigators and folks who work at the city libraries, and they’re really trusted individuals. Folks in those spaces are safe people to talk to; they don’t have to be health professionals.*


Increasing vaccine confidence could be improved by working with people who were representative of the communities served. The following quotes illustrate the importance placed by stakeholders in hiring and working with immigrant and refugee community members:


*One of the things that we realized was important is that we had to have representation. When you’re actually being educated and served by people who look like you, there’s better uptake, they can understand our perspective and it’s not something to be scared of.*


Partnerships between primary care stakeholders and communities agencies were viewed as essential to vaccine provision and promotion efforts. For example, several primary care stakeholders described community partnerships that existed pre-COVID that could be leveraged at this time (e.g., community development framework, shelters, and schools). Other partnerships were newer. For example, Black community leaders and health organizations were mobilized to address unmet COVID-19 vaccination needs in the wake of public health efforts that did not prioritize their communities.


*It would be naive of us to think that we could do it without engaging with our partners. You need to reach out to those key individuals and organizations to help mobilize a response in the community.*


Stakeholders also spoke about the importance of ensuring the provision of holistic, flexible, and low-barrier services. The provision of services addressing the SDOH was especially effective for immigrant and refugee community members who face access barriers and/or have other priorities not specific to COVID-19 and/or who have had previous negative experiences with health providers. As one primary care stakeholder explained,


*What our team realized… was that they really needed to create supports to address both the social determinants of health as well as the trauma and to be able to approach trauma by decolonizing, the Western medical system, and creating an opportunity to also embrace Indigenous approaches to health and wellness.*


Primary care stakeholders recognized the effectiveness of translating COVID-19 messaging to meet the cultural and linguistic needs of immigrant and refugee communities and tailoring those messages, since core public health messaging did not always reflect people’s experiences.


*There was a strategy to ensure that multilingual outreach staff were available to provide access to up-to-date credible information in languages that the receiving clients could understand.*


## 4. Discussion

Our study identifies the barriers to equitable vaccine promotion and provision among equity-deserving communities, and, more specifically, immigrant and refugee communities, in Canada, including distrust, misinformation, lack of confidence in vaccines, and access/system-level barriers. The barriers identified in our research confirm the results of previous Canadian research on vaccine hesitancy, including lack of trust in vaccines and in the healthcare system and lower health literacy [30,37].

It is well-documented that immigrant and refugee populations experience multiple and intersecting barriers to accessing appropriate and responsive health services, including institutional discrimination, lack of awareness of available services, lack of culturally appropriate services in relevant languages, and lack of culturally competent health professionals [38,39]. Numerous scholars have identified the urgent need to build capacity and reduce stigma and paternalism among health providers working with immigrant and refugee communities [40,41].

Our findings are consistent with international research on barriers to COVID-19 vaccine promotion and provision among immigrant and refugee populations. The WHO’s ‘COVID-19 Immunization in Refugees and Migrants: Principles and Key Considerations’, and other international studies identified similar factors associated with vaccine hesitancy, including distrust, social exclusion, and other systemic barriers. Many refugees and migrants face language barriers, lack culturally appropriate health information, and are exposed to misinformation [23]. The importance of communication and effective messaging in building trust and countering misinformation, fake news, and misconceptions among immigrant and refugee populations has been emphasized [42]. Concerns about vaccine safety, efficacy, and cost—particularly when vaccines are not freely available—further hinder vaccine acceptance.

Our findings further suggest that many of these barriers could be effectively reduced by adopting relationship-building and equity-driven approaches, such as community engagement and development, focusing on the social determinants of health, and selecting COVID-19 best-practices, such as the provision of one-on-one supports, culturally and linguistically effective communication, and working with community members as valued partners and staff.

This study specifically highlights the importance of using diverse tailored and targeted approaches when engaging in vaccine provision and promotion with immigrant and refugee populations. Evidence suggests that one size does not fit all in terms of the responding to public health disasters in priority populations [43,44]. Individuals are more likely to access primary healthcare and social services in a welcoming and inclusive setting where they have trusted interpersonal relationships with their healthcare providers. Clients want and need health services that address their priority needs, as well as essential health or wraparound services [45,46].

Social determinants of health, including structural inequalities and discrimination, provide a useful framework to account for the disproportionate health risks and differential health outcomes experienced in equity-deserving populations [47,48,49]. In the case of COVID-19, excess cases and deaths have been attributed to structural factors related to income, employment, food insecurity, and the built environment which necessitate, for example, risky working conditions outside the home and using public transportation [50,51]. It is also important to recognize that exposures from childhood, youth, and mid-life affect health in adult and later life, as well as health across generations [52]. For migrant health in particular, the life course approach highlights the need to consider social determinants from pre-migration through to settlement, and even to the next generation [53]. Education, occupation, and health literacy pre-migration determine access to vaccine information and confidence in Canada and subsequent COVID-19 outcomes.

It is notable that several of the identified facilitators to vaccination are exemplified in the health service delivery model of Canada’s Community Health Centers (CHCs). A recent survey estimated that 59% of clients served by CHCs are recent immigrants and refugees (this proportion rises to 68% in Ontario [54]), 55% of clients are racialized, and 87% may be classified as low income [55]. Throughout the COVID-19 pandemic, CHCs engaged in a wide range of COVID-19 vaccine promotion and provision services to reach individuals from equity-deserving populations. Using a relational, holistic, and comprehensive approach, CHCs addressed the social determinants of health and vaccine inequities, including housing insecurity, poverty, language barriers, and ensured low-barrier access to primary care services. CHCs used innovative targeted approaches to disseminate vaccine information with clients and community members and implemented tailored strategies to address the informational needs of their equity-deserving populations they serve. CHCs use of community engagement and development strategies were effective in building trust and ensuring that vaccine services were responsive to community-defined needs.

The disproportionate impact of COVID-19 on immigrants and refugees is not limited to the risk and rate of COVID-19. Unless equity-informed recovery models are put in place which prioritize appropriate primary healthcare for these groups, health inequities will persist and create greater disparities among the Canadian population. The barriers identified in this study are not unique to the provision of vaccine services; these factors also affect primary healthcare access. Primary care providers, including family physicians, need to consider barriers and facilitators to health services, which directly affect how patients from equity-deserving populations receive care. By addressing the barriers and needs of equity-deserving populations, primary care providers can facilitate the adoption of approaches that may result in a more equitable healthcare system, serving the needs of all individuals and communities across Canada.

### Limitations

Several study limitations must be acknowledged. Firstly, the data presented are from interviews with primary care stakeholders drawn primarily from CHCs and may not be representative of primary care stakeholders in Canada as a whole. Within CHCs, there is variation in terms of location, size, staff composition, range of services/programs offered, and populations served. For example, some CHCs work primarily with immigrant and refugee populations, whereas others tailor their programming to specific racial or ethnic communities.

Second, both pre-migration factors and post-migration social determinants of health influence healthcare access among diverse immigrant and refugee communities. Primary care respondents did not always differentiate between the barriers and facilitators experienced by immigrants versus refugees, instead emphasizing shared experiences. Additionally, some primary care stakeholders made broad generalizations about ethnic or racialized groups without acknowledging the diversity within these communities. While this diversity is well recognized [56], commonalities in COVID-19 vaccine experiences have also been observed [57].

Thirdly, the findings presented on the barriers and facilitators to vaccine promotion and provision among immigrants and refugees represent primary care stakeholder perspectives and may not be representative or inclusive of the experiences of immigrants and refugees across Canada. Fewer studies examined the barriers and facilitators to COVID-19 vaccination from the perspectives of equity-deserving populations [57,58].

## 5. Conclusions

The risk and impacts of COVID-19 are disproportionately distributed in Canada and worldwide. In order to reduce health inequities, it is critical to identify and address barriers and facilitators to vaccine promotion and provision in equity-deserving populations, including immigrants and refugees. High levels of distrust, misinformation, lack of vaccine confidence, and systemic barriers are most effectively addressed using people-centered community-based holistic approaches. Since 2021, CHCs across Canada have played a significant role in promoting access and confidence in COVID-19 vaccines. Their success in the vaccine rollout highlights the urgent need for policymakers to provide ongoing support to enhance and sustain the work of CHCs and promote and expand this model of health service delivery. The CHC model’s relational and equity-focused approach offers an effective framework for reducing health inequities, making it relevant and adaptable to health systems globally.

## Figures and Tables

**Table 1 ijerph-22-00493-t001:** Barriers to Vaccine Promotion, Provision, and Uptake among Immigrant and Refugee Populations.

**Historical distrust of government and health services** **Increased rates of misinformation** ○ Lack of availability of reliable, culturally appropriate and evidence-based information ○ Low perception about disease severity ○ Influence of the anti-vaccine movement **Health messaging gaps** ○ Low literacy levels ○ Low/no digital literacy ○ Difficulty reaching vulnerable communities **Lack of vaccine confidence** ○ Doubts about vaccine safety ○ Fear of adverse events/side effects of vaccines ○ COVID-19 and vaccine fatigue **Systemic barriers** ○ Lack of access to primary care in the population ○ Failure of health system to understand/address SDOH impacting on COVID-19 in equity-deserving communities ○ Lack of access to vaccines at CHCs **Complex client needs ** ○ Dealing with trauma and cultural safety needs ○ Addressing the social determinants of health **Access barriers** ○ Distance to vaccine clinics ○ Access to transportation ○ Availability of vaccination during non-working hours

**Table 2 ijerph-22-00493-t002:** Facilitators to Vaccine Promotion and Provision among Equity-Deserving Populations.

**Trusted relationships with healthcare providers** ○ Relationship building between healthcare providers and clients ○ Addressing SDOH and cultural considerations before vaccination **Community engagement and strength-based approaches** ○ Fostering community engagement through targeted outreach ○ Capacity building of communities served ○ Utilizing peer ambassadors and community connectors ○ Championing resident leaders within communities **Interdisciplinary collaboration** ○ Partnerships with social, community, and health organizations to expand vaccine access **Culturally tailored and appropriate resources/messaging** ○ Translation of evidence-based information in diverse and priority languages ○ Culturally and linguistically accessible messaging ○ Availability of information for individuals with low and no literacy ○ Integrated vaccine discussions into primary care and one-on-one client interactions. **Accessible and convenient vaccine services** ○ Vaccine services available at the CHC ○ Community-based access through mobile clinics and events ○ Culturally responsive, and decolonized approaches ○ Walk-in, on-site, and off-site vaccine delivery options **Targeted vaccine communication** ○ Leveraging of WhatsApp, video, and radio for reliable messaging ○ Empowerment of local leaders to disseminate vaccine messaging

## Data Availability

The data for this study are available through the Canadian Association of Community Health Centers.

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
