# Peer review of "Factors Impacting COVID-19 Vaccine Uptake and Confidence Among Immigrant and Refugee Populations in Canada"

_ijerph, 2025, doi:10.3390/ijerph22040493_

Round 1
Reviewer 1 Report (Previous Reviewer 1)
Comments and Suggestions for Authors
The authors reviewed the manuscript, following most of the suggestions. I recommend publishing the paper. Some typing errors should be corrected.
Author Response
Comments: The authors reviewed the manuscript, following most of the suggestions. I recommend publishing the paper. Some typing errors should be corrected.
Response: The manuscript was reviewed for typing errors.
Reviewer 2 Report (New Reviewer)
Comments and Suggestions for Authors
This article presents a discussion on access to vaccination for refugees and immigrants in Canada, with the aim of understanding the obstacles that exist for this population. The article is consistent, the methodology is presented clearly, and the discussion presents a relevant contribution to the field of studies. In the discussion regarding the social determinants of health, it would be possible to broaden the reflection by considering that the determinants act on immigrants and refugees even before they enter the migratory circuit. In this sense, the conceptions about health and illness, as well as the validity of vaccination, also accompany immigrants and refugees from their home countries, which may have some impact on whether or not they seek vaccination. In any case, the article presents an important discussion and the ethical principles were clarified in the manuscript. Therefore, this reviewer is in favor of publication.
Author Response
Comments: In the discussion regarding the social determinants of health, it would be possible to broaden the reflection by considering that the determinants act on immigrants and refugees even before they enter the migratory circuit. In this sense, the conceptions about health and illness, as well as the validity of vaccination, also accompany immigrants and refugees from their home countries, which may have some impact on whether or not they seek vaccination.
Response: In the revised version we add a discussion about the importance of considering pre-migration social determinants of health.
It is also important to recognize that exposures from childhood, youth, and mid-life affect health in adult and later life as well as health across generations (Kuh & Ben-Shlomo 2004). For migrant health in particular, a life course approach highlights the critical importance of considering social determinants from pre-migration through to settlement, and even to the next generation (OECD, 2020). Education, occupation and health literacy pre-migration undoubtedly affect access to vaccine information and confidence in Canada.
Kuh, D. and Y. Ben-Shlomo (2004). Introduction. Chronic Disease Epidemiology. D. Kuh and Y. Ben-Shlomo. Oxford, Oxford University Press.
OECD (2020). Migration Policy Debates © OECD N°25, November 2020.
Reviewer 3 Report (Previous Reviewer 2)
Comments and Suggestions for Authors
The new additions to the paper appear to make it ready for publication. There were four minor wording/text errors and these are listed below for ease of final text editing. They do not appear to be a reason to hold up the overall publication of the paper.
Pg 8, line 288 there appears to be an extra space between the word 'and' and 'other' that should be removed.
Pg 8, line 289, a full stop needs to placed after the word barriers.
Pg 8, line 290, a full stop should be placed after the word misinformation.
Pg 9, line 313, for readability suggest the word 'to' be inserted so sentence reads "...have been attributed to structural..."
Author Response
Comments: The new additions to the paper appear to make it ready for publication. There were four minor wording/text errors and these are listed below for ease of final text editing. They do not appear to be a reason to hold up the overall publication of the paper.
Pg 8, line 288 there appears to be an extra space between the word 'and' and 'other' that should be removed.
Pg 8, line 289, a full stop needs to placed after the word barriers.
Pg 8, line 290, a full stop should be placed after the word misinformation.
Pg 9, line 313, for readability suggest the word 'to' be inserted so sentence reads "...have been attributed to structural..."
Response: The above corrections were made in the revised submisison.
This manuscript is a resubmission of an earlier submission. The following is a list of the peer review reports and author responses from that submission.
Round 1
Reviewer 1 Report
Comments and Suggestions for Authors
The study focuses on data from primary care stakeholders through a survey and in-depth interviews. Below are some considerations to make the manuscript more theoretically and methodologically dense.
In general, the text presents little dialog with international literature. It would be interesting to present a brief review of how these problems identified by stakeholders (of barriers and facilitators) have occurred in other contexts outside Canada during the pandemic. This experience is certainly not exclusive to Canada.
The introduction cites data on immigrants and refugees, and throughout the text, quotes and texts on ethnic or racial minorities are presented.
The aim is to focus on stakeholders' representations of vaccination among immigrants and refugees in Canada.
Some issues deserve reflection:
1. It is problematic to treat migrants and refugees as homogeneous groups. The literature on international migration discusses the fragility of these views, which end up stereotyping and reifying conceptions of such diverse populations. Beware of references such as "African," "Black," and "Caribbean." Africa is as diverse as any other country. No cultural, socio-economic, or political identity exists in these expressions. If it is necessary to use them, I suggest a critique in the footer.
2. The text leaves the impression that there are, on the one hand, the holders of scientific knowledge of the biomedical model and, on the other, ethnic-racial minorities and immigrants and refugees. International literature, especially in medical anthropology, is critical of these closed dichotomies as if, on one side, there is a scientific "truth" and, on the other, "beliefs."
3. As there is no description in the text of the migrant and refugee populations who access health services, the data is abstract and decontextualized. For example, migrants and refugees with higher education probably share scientific conceptions about the vaccine. The time migrants have lived in Canada, their country of origin, and the socio-economic conditions in which they arrived vary greatly. Therefore, any interpretation by stakeholders of these populations runs the risk of being stereotyped
4. In the case of vaccination barriers, it would be essential to address the issue of the transnationality of relations between immigrants and refugees. Whether or not to take vaccines is related to information from transnational networks, such as family members in other countries. In the case of disinformation, it is essential to discuss the fake news spread globally about vaccines and anti-vaccine movements. Regarding disinformation, it seems that it is associated with ignorance or lack of scientific knowledge on the part of immigrants and refugees. It's vital to problematize this issue. Immigrants who come from countries where there is great distrust of vaccines (due to unethical testing) or do not have solid vaccination programs may be hesitant to vaccinate.
2. Materials and Methods
The survey data is not presented, not even the subject of the questions asked.
It would be essential to describe the number of CHCs in Canada and which locations participated in the survey. Are they urban locations? Rural locations? How is access to health care facilities? Do they represent places with a more significant presence of migrants and refugees?
There are no results on this survey to understand how they arrived at the in-depth interviews conducted with organizations that identified 'best practices' to further explore barriers and facilitators to implementing the identified 'best practices.'
The thematic analysis does not cite any theoretical reference that guided it, making the study descriptive. The intersectional analysis also does not cite bibliographical references—the same for SDoH.
Results and discussion
The way the survey responses relate to the interviews in the results is unclear.
Defining the equity-deserving populations in Canadian public policies is essential, especially in health.
A theoretical problematization or dialogue with the literature is essential to analyze the barriers and facilitators mentioned.
I suggest thoroughly revising the text to critically problematize the data obtained in the survey and in-depth interviews. Dialogue with international literature will also help to improve the discussion of the results.
Reviewer 2 Report
Comments and Suggestions for Authors
This is an interesting paper and is an important contribution to how a better job can be done by health agencies to reach and vaccinate immigrant and refugee populations in Canada. The results could also be applied to other similar populations in other countries.
As noted in the scoring the methodology and discussion could be improved upon. Firstly, in terms of the methodology a number of questions about the questionnaire are unanswered. For example, how many questions did it contain, what was asked about, how long was it and how long did it take to complete? What was the response rate and were all questionnaires received used in the study? For example, were some discarded due to being incomplete? It is normally standard practice to supply this type of information.
In terms of the interviews, information about how the interviewees were selected and the length of the interviews would normally be supplied in the methods section. Furthermore, it is said a thematic analysis was conducted which is fine and yet was a particular method used to arrive at the themes? For example, was Braun & Clark's (2023) thematic analysis method followed or something else. Finally, was a test of intercoder reliability conducted and however were differences negotiated between the coders?
This leads to the discussion. It is unusual to see the results from a questionnaire and interviews being blended together in the manner they are currently written. Normal practice is to discuss the results of the two analyses separately so the reader understands where the information comes from. Can this be done here? Also could a copy of the questionnaire be supplied as an appendix?
Finally, there is no mention of an application for or the granting of ethical approval for this study. This should be included too.
This might seem a lot and yet it should not take too long as the bulk of the work is done. There are a few minor wording errors and yet these can be examined after these more substantive areas have been examined.